# Improved Graft Function following Desensitization of Anti-AT_1_R and Autoantibodies in a Heart Transplant Recipient Negative for Donor-Specific Antibodies with Antibody-Mediated Rejection: A Case Report

**DOI:** 10.3390/ijms25042218

**Published:** 2024-02-12

**Authors:** Regina Jung, Kevin Ly, Michiko Taniguchi, Aileen Grace Arriola, Christopher Gravante, Derek Shinn, Leena Mathew, Eman Hamad, Steven Geier, Abdelhamid Liacini

**Affiliations:** 1Department of Pharmacy, Lahey Hospital and Medical Center, Burlington, MA 01805, USA; 2Department of Pharmacy, Temple University Hospital, Philadelphia, PA 19140, USA; 3Immunogenetics Laboratory, Temple University Hospital, Philadelphia, PA 19140, USA; 4Department of Pathology and Laboratory Medicine, Lewis Katz School of Medicine, Philadelphia, PA 19140, USA; 5Heart and Vascular Institute, Section of Cardiology, Lewis Katz School of Medicine, Philadelphia, PA 19140, USA

**Keywords:** orthotopic heart transplantation, autoantibodies, rejection, antibody-mediated rejection, AT_1_R

## Abstract

HLA donor-specific antibodies (DSAs) pre and post transplant increase the risk of antibody-mediated rejection (AMR) and lead to poor graft survival. Increasing data exist to support the involvement of non-HLA antibodies in triggering an immunological response. The development of non-HLA antibodies specific for AT_1_R is associated with poor clinical outcomes in orthotopic heart transplant recipients. This case presents an investigation of non-HLA antibodies in a 56-year-old female heart transplant recipient diagnosed with AMR in the absence of DSAs.

## 1. Introduction

The presence of HLA donor-specific antibodies (DSAs) pre and post transplantation increases the risk of antibody-mediated rejection (AMR) and is associated with poor graft survival. There are also increasing data to support that the presence of non-HLA antibodies, such as angiotensin II type-1 receptor (AT_1_R), MHC class I polypeptide-related sequence A/B (MICA/B), autoantibodies, and anti-endothelial cell antibodies (AECAs), can trigger an immunological response in the absence of DSAs. AT_1_R is present on the endothelial cells’ surface and regulates blood pressure and fluid balance through the renin–angiotensin–aldosterone system (RAAS). The activity of angiotensin II on AT_1_R also stimulates vascular smooth muscle proliferation, cellular migration, fibrosis, and inflammation. Consequently, anti-AT_1_R antibodies can cause endothelial cell activation and have proinflammatory and profibrotic effects that have been implicated in reports of AMR in cardiac and other transplant recipients [1]. Known risk factors for the formation of anti-AT_1_R antibodies include blood transfusions, pregnancy, and a history of mechanical circulatory support commonly used as a bridge to heart transplantation [2]. We report a case of a heart transplant recipient presenting with anti-AT_1_R antibody-induced AMR, highlighting the need for pre-transplant testing for non-HLA antibodies in select patients to identify those at risk of AMR despite the absence of DSAs.

## 2. Case Presentation

A highly sensitized (cPRA 90%) 56-year-old female patient received an orthotopic heart transplant (OHT) due to her history of end-stage congestive heart failure and cardiomyopathy. The pre-transplant HLA antibody testing showed a borderline presence of class I DSA (Figure 1). The pre-transplant serum was strongly positive for anti-AT_1_R (titer 1:6400) and five other autoantibodies (>50% above the respective cutoffs) (Figure 1A,B). 

Notably, there was concern for early AMR based on a biopsy within one month of her transplant, which showed diffuse C4d staining (>50%) and a strong positivity. This was concurrent with the emergence of a de novo class I DSA that responded well to a course of oral corticosteroids. The patient was asymptomatic without clinical evidence of allograft dysfunction based on an assessment via cardiac catheterization and echocardiogram. Subsequent HLA antibody testing a month later indicated a reduction in DSAs below the threshold. On day 255 post transplantation, the recipient underwent a routine biopsy that, once again, suggested AMR, with diffuse C4d staining (ISHLT 2013 pAMR 2 [H1, I1]) (Figure 2). The patient was again asymptomatic without clinical evidence of allograft dysfunction but treated with five sessions of plasmapheresis and IVIG, in addition to increasing the dose of corticosteroids (Table 1). Subsequent tests for DSAs remained negative. However, the patient had a persistently high level of AT_1_R antibodies (>40 U/mL) and autoantibodies against EIF2A (eukaryotic translation initiation factor 2A), PRKCZ (protein kinase C zeta type), PTPRN (receptor-type tyrosine- protein phosphatase-like N), PRKCH (protein kinase C eta type), and GDNF (glial cell line-derived neurotrophic factor). Routine biopsy on day 481 post transplantation once again found evidence of AMR with positive C4d and C1q immunofluorescence in the presence of high-level anti-AT_1_R and autoantibodies with no class I and class II DSAs. Cardiac catheterization at this time revealed normal hemodynamics but a presence of cardiac allograft vasculopathy (CAV), as evidenced by 30% stenosis of the proximal left anterior descending artery and 30% stenosis of the first diagonal branch. Based on the absence of DSAs and presence of biopsy-proven AMR, the patient was determined to have recurrent AMR related to non-HLA antibodies. Treatment was initiated with a second set of five plasmapheresis sessions, IVIG, a steroid taper, and rituximab (Table 1). The patient was also started on valsartan 40 mg daily at this time. The patient remained strongly positive for AT_1_R antibodies even after treatment, and the decision was made on day 646 post transplantation to attempt controlling her persistent AMR with monthly extracorporeal photopheresis (ECP), which has continued to the date of writing.

## 3. Discussion

AMR in the setting of OHT has a major impact on patient survival, but management of this condition still lacks full consensus among experts [3,4]. The presence of DSA is associated with increased risk of graft dysfunction and development of AMR and increased risk of graft loss [5]. The impact of non-HLA antibodies on patient outcomes is comparatively less well established. However, data gathered in renal transplant recipients suggest a significant role for non-HLA antibodies in immune responses directed at the allograft that may result in endothelial cell apoptosis [6]. The role of non-HLA antibody involvement in allograft rejection is further supported by a report of rejection occurring in renal transplantation conducted between HLA-identical siblings to which undetectable minor HLA antibodies may have contributed [7]. 

Of the non-HLA antibodies which have been studied in association with clinical outcomes, the most prominent are those that are directed at AT_1_R. Endothelial cell activation can have proinflammatory and profibrotic effects that may be implicated in AMR and impact graft function. The prevalence of AT_1_R antibodies in OHT recipients ranges from 40 to 60% and may be associated with increased risk of AMR, cellular-mediated rejection (CMR), and CAV [1,8]. These effects have been observed in synergy with anti-HLA antibodies, suggesting a greater effect when AT_1_R- and HLA-directed antibodies are present in tandem [9,10]. The presence of strong pre-transplant AT_1_R antibodies has been associated with a higher risk of graft loss and early AMR after transplant. Additional reports in kidney transplant recipients have described accelerated rejection and worse graft survival in the presence of AT_1_R and HLA antibodies [10]. The effect of AT_1_R antibodies on the graft also does not appear to be complement-dependent, with multiple reports of biopsy-proven, C4d-negative acute AMR [11,12]. A definitive understanding of the contribution of AT_1_R antibodies on post-OHT survival and AMR remains controversial due to conflicting reports on their impact [2,13]. The prevalence of AT_1_R-associated AMR is unknown in heart transplant recipients, but a large review including 1845 kidney transplant recipients described a 21% incidence of pathologic features of active AMR with or without DSAs in the 504 patients who were positive for the AT_1_R antibody [14].

While a guideline-directed standard for the management of AMR does not exist at this time, the treatment plans (Table 1) for this patient followed standard protocols at our institution that are based on common practice previously described for the management of AMR in this patient population [2,15,16,17]. Nonetheless, the patient remained strongly positive for AT_1_R antibodies and certain autoantibodies. We continue to manage her persistent AMR with monthly ECP, a cellular immunotherapy involving the collection and centrifugation of blood to isolate white blood cells which are then exposed to photodynamic therapy. The true therapeutic mechanism remains unknown but is theorized to involve the induction of lymphocyte apoptosis [18]. While there are multiple studies demonstrating mixed response rates and outcomes in patients with recurrent rejection, our patient remains free of recurrent rejection with no progression of her mild CAV after ten months of therapy to date [19,20,21]. The 2010 ISHLT guidelines for the care of heart transplant recipients currently mention ECP as a potential treatment option for recurrent or resistant acute cellular rejection (Class 2, level of evidence B) but do not provide a specific recommendation regarding AMR [22]. The patient in our case study was also started on a low dose of valsartan (40 mg twice daily) late into therapy (494 days post transplant), as angiotensin receptor blocker (ARB) therapy has displayed a theoretical benefit in the prevention of vasculopathy through the blockage of antibody-mediated AT_1_R activation [13]. 

This case suggests a more routine role for the pre- and post-transplant monitoring of non-HLA antibodies, as testing for AT_1_R antibodies was not conducted until day 255 post transplant once suspicion formed due to the presence of AMR in the absence of DSA. Routine testing for more common non-HLA antibodies such as AT_1_R may present a benefit, as early detection may guide therapeutic decision making such as the timely initiation of ARB therapy. However, as the treatment for HLA-mediated rejection is not significantly different from that of non-HLA-mediated rejection, the full extent of this benefit is yet to be explored. This patient remains strongly positive for AT_1_R antibodies, and there remains a lack of guidance on what antibody trend represents an appropriate response to therapy. Treatment modalities for AMR such as plasmapheresis, steroids, and rituximab are not without significant side effects, and more information is required to accurately guide patient exposure to these therapies. The patient remains active with no cardiac complaints at the time of writing despite her persistently high AT_1_R antibody titer requiring continuing therapy. Moving forward, we suggest a management strategy for patients with the aforementioned risk factors for AT_1_R antibody development involving the inclusion of AT_1_R antibodies on the screening of pre-transplant serum and periodic testing for de novo antibodies. Patients who screen positive for AT_1_R antibodies should be initiated on ARB therapy as soon as is safely able and monitored appropriately.

There are multiple limitations that should be considered when reviewing this case report. Our laboratory reported consistently high AT_1_R antibodies (>40 units/mL), but the clinical impact of these high antibody levels remains unclear in the current body of heart transplantation literature. We observed, in this case, that the persistence of these values well-above our laboratory’s cutoff for positivity (>17 units/mL) was not associated with dysfunction of the allograft but was present in tandem with histologic markers of AMR such as diffuse C4d staining. Because we did not initiate valsartan until this patient’s second course of treatment for AMR, we are unable to draw conclusions as to how an earlier initiation of ARB therapy would have impacted her care. Data regarding the impact of ARB therapy on AMR associated with AT_1_R antibodies and optimal agent selection are limited and warrant further investigation. We acknowledge the difficulty in quantifying the contribution of autoantibodies other than those specific to AT_1_R to this patient’s histologic presentation of rejection and note that her development of mild cardiac allograft vasculopathy is likely the result of synergism between HLA- and non-HLA-related antibodies.

Our case supports previous reports of synergistic contribution between HLA- and non-HLA antibodies for AMR. Non-HLA AMR may present with or without intravascular C4d staining, but this patient’s biopsies consistently presented with C4d immunofluorescence even in the absence of DSA. Therefore, non-HLA antibody testing cannot be limited to only cases of AMR absent of or with weak C4d deposition.

## 4. Conclusions

The present case demonstrated persistent AMR that was initially accompanied by low-level DSAs that later resolved. Concurrently, the patient tested strongly positive for anti-AT_1_R and autoantibodies pre and post transplantation, highlighting the potential value of more thorough pre-transplant testing for non-HLA antibodies to guide early treatment and close antibody monitoring. An optimized schedule for post-transplant monitoring and management in the setting of non-HLA AMR remains an area for exploration that may benefit from the development of a protocolized approach.

## Figures and Tables

**Figure 1 ijms-25-02218-f001:**
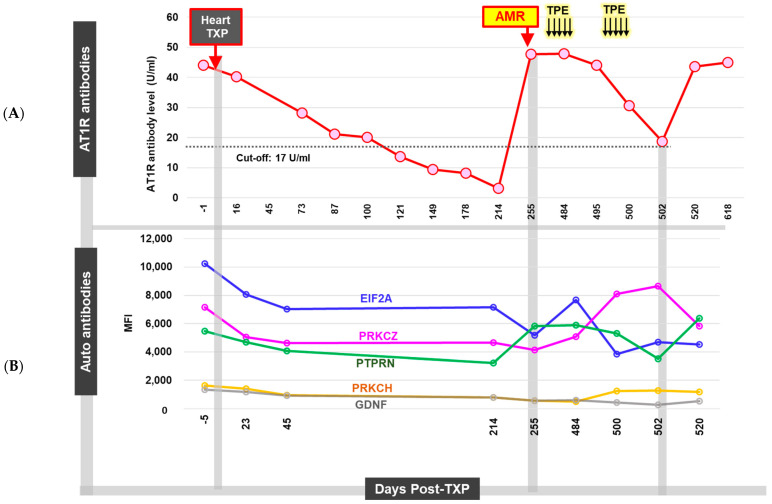
Non-HLA (AT1R and autoantibodies) monitoring during AMR diagnosis. (**A**) Following the biopsy-proven AMR diagnosis, the patient underwent five initial sessions of therapeutic plasmapheresis (TPE), followed by an additional five sessions of TPE. On day 502, the anti-AT1R level temporarily fell by 61% from 47.8 to 18.7 U/mL. The patient was discharged. However, the anti-AT1R level started to increase again and, at the time of writing, were at a high level (>40 U/mL). (**B**) After the AMR diagnosis, the levels of the five non-HLA antibodies remained above the threshold. (Abbreviations: EIF2A, eukaryotic translation initiation factor 2A; PRKCZ, protein kinase C zeta type; PTPRN, receptor-type tyrosine-protein phosphatase-like N; PRKCH, protein kinase C eta type; and GDNF, glial cell line-derived neurotrophic factor.

**Figure 2 ijms-25-02218-f002:**
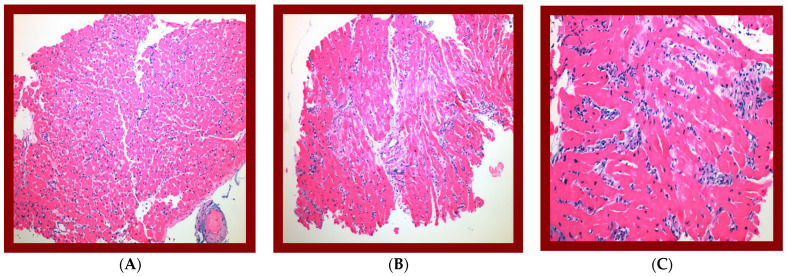
Biopsy evaluations have shown histological suspicious for antibody-mediated rejection at POD 180:—ISHLT (2004) grade 1R—focus of lymphocytic infiltrate with myocyte damage—2013 ISHLT. Presented as mononuclear inflammatory response infiltrating myocardial tissue with predominant lymphocytic cells associated with myocyte damage. Immunohistologic assessment confirmed 2013 ISHLT pAMR 1 (H+); histologic findings suspicious for AMR with negative immunopathologic findings. Immunofluorescence assessment confirmed negative staining for C4d deposition and C1q. (**A**) 100×. H&E of one biopsy fragment showing interstitial lymphocytic infiltrate with myocyte damage, compatible with grade-1R acute cellular rejection. (**B**) 100×. H&E of separate biopsy fragment showing increased interstitial cellularity, suspicious for antibody-mediated rejection. (**C**) 200×. Higher power of increased interstitial cellularity showing prominent endothelial cells and interstitial edema with basophilic staining, confirming impression of AMR.

**Table 1 ijms-25-02218-t001:** AMR treatment plan at post-transplant day 255 and day 481.

Day Post Transplantation	TPE	Steroids	IVIG	Rituximab
255		Methylprednisolone 1 g		
256	✓	Methylprednisolone 1 g	250 mg/kg	
257		Methylprednisolone 1 g		
258	✓	Prednisone 50 mg twice daily	250 mg/kg	
259		Prednisone 50 mg twice daily		
260	✓	Prednisone 40 mg	250 mg/kg	
261		Prednisone 40 mg		
262	✓	Prednisone 30 mg	250 mg/kg	
481		Methylprednisolone 1 g		
482	✓	Methylprednisolone 1 g	250 mg/kg	
483		Methylprednisolone 1 g		
484	✓	Prednisone 40 mg	250 mg/kg	
485		Prednisone 40 mg		
486	✓	Prednisone 30 mg	250 mg/kg	
487		Prednisone 30 mg		
488	✓	Prednisone 20 mg	250 mg/kg	
489		Prednisone 20 mg		
490	✓	Prednisone 10 mg	1 g/kg	
491		Prednisone 5 mg daily thereafter		375 mg/m^2^

✓ = TPE received on this date.

## Data Availability

Data are contained within the article.

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
