# Peer review of "Improved Graft Function following Desensitization of Anti-AT1R and Autoantibodies in a Heart Transplant Recipient Negative for Donor-Specific Antibodies with Antibody-Mediated Rejection: A Case Report"

_ijms, 2024, doi:10.3390/ijms25042218_

Round 1
Reviewer 1 Report
Comments and Suggestions for Authors
Manuscript title: "Improved Graft Function Following Desensitization of Anti-AT1R And Autoantibodies In A DSA-Negative Heart Transplant Recipient with AMR: A Case Report"
The manuscript addresses an important and niche aspect of heart transplant rejection, focusing on non-HLA antibodies, especially AT1R antibodies. This is a valuable contribution to the field given the rarity of such cases.
Suggestions for reflection and improvement:
1. Introduction - A more comprehensive review of the existing literature on non-HLA antibodies in heart transplant rejection would strengthen the introduction. This would provide a clearer context for the study's significance.
2. The methodology section could benefit from more detailed descriptions of procedures and justifications for the chosen treatment approach. Comparisons with alternative or newer methodologies will be provided. At the same time, including more detailed charts or tables illustrating the patient's response to treatment over time would increase readers' understanding.
3. Although the authors describe their case quite well, in my opinion, it would be useful to expand the description of the study's limitations, such as the possibility of generalization based on a single case, which was not discussed in detail. A more detailed discussion of the limitations and implications of the findings is necessary. At the same time, authors may consider adapting the language for a broader audience by simplifying the technical language without losing the scientific rigor of the report.
4. As for the bibliography section, as I mentioned above, it would be useful to conduct a broader literature review.
To sum up
This case report provides valuable information on the role of non-HLA antibodies in heart transplant rejection, which is a relatively understudied area of transplant immunology. The manuscript is well constructed and makes the case clearly. However, a more detailed discussion of its limitations, a deeper analysis of the clinical implications, and a more comprehensive literature context would be useful. Including newer research and improvements in data visualization would greatly strengthen the report.
​
Author Response
Thank you for your helpful feedback. Please see below for a point-by-point response:
- A more comprehensive review of the literature (particularly surrounding non-HLA antibodies in heart transplant, as well as other forms of transplant) has been included with appropriately updated references.
-
We have included a justification for the AMR treatment plan chosen for this patient in the updated copy of the manuscript. At this time, there is not a standardized, guideline-directed approach for treatment of AMR which makes comparison of alternative or newer methodologies difficult.
-
A paragraph describing some limitations to conclusions which can be drawn from this case report has been included in the discussion.
-
Please see response to point 1.
Reviewer 2 Report
Comments and Suggestions for Authors
In this manuscript the authors presented a case demonstrated persistent AMR which was initially accompanied by low-level DSAs that later resolved. Concurrently, the patient tested strongly positive for anti-AT1R and autoantibodies pre- and post-transplantation, highlighting the potential value of more thorough pre-transplant testing for non-HLA antibodies to guide early treatment and close antibody monitoring.
As the authors insist an optimized schedule for post-transplant monitoring and management in the setting of non-HLA AMR is required.
I have some comments
1.In this paper, there is no data regarding the patient's cardiac function after AMR occurs. Comparative data on changes in AT1R and cardiac function is needed.
2. Regarding the histological image in Figure 2, are there any C4d-positive findings in CD68-positive macrophages or capillary endothelial cells? Also, are there any findings such as deposition of Immunoglobulin, C3d or C1q? Please also add these immunological characteristics.
3.What do you think of the therapeutic effects of monthly photopheresis? Is this treatment effective in reducing AT1R? How long have you been on treatment?
Author Response
Thank you for your helpful feedback. Please see below for a point-by-point response:
- There were no changes in actual cardiac function. The patient’s asymptomatic status was clarified with multiple additions to the manuscript. Mild CAV did develop, which was added to the manuscript with a description of timing.
-
Yes, thank you very much for pointing out the Immunohistologic assessment results in Figure 2. As you know HLA and Non-HLA antigens are not expressed equally among all individuals; this increases the potential of such proteins to act as non-self-antigens and activate cytotoxic and helper T cells. T cells respond to these donor antigens either directly or indirectly based on the method of antigen presentation. T cells can either directly recognize donor alloantigens on allograft or when presented indirectly or semidirectly by recipient antigen-presenting cells. The antibody reacts to donor MHC antigens and non-MHC antigens (HLA-Class I and Class II, AT1R, Coll, Vimentin, LG3, PP1A, PLAR2R, LMNA,…) leading to capillary endothelial changes. The deposition of complements fragments within myocardial capillary are detectable by immunofluorescence.
In our hand patient demonstrated ISHLT (2004) grade 1R, presents as a mononuclear inflammatory response infiltrating myocardial tissue with predominant lymphocytic cells with myocyte damage. Immunohistologic assessment can confirm 2013 ISHLT p AMR 1 (H+); histologic findings suspicious for antibody mediated rejection, with negative immunopathologic findings. Immunofluorescence assessment confirmed negative immunofluorescent staining for C4d deposition and C1q.
Included a summarized version of the above justification in the updated copy of the manuscript.
- A more detailed explanation of the patient’s monthly ECP and the place of this therapy in treatment has been included in the updated copy of the manuscript.
Reviewer 3 Report
Comments and Suggestions for Authors
The case in very interesting and important for the transplantation management. I would suggest to describe a more detailed pretransplantation and postHTX treatment and detection plan for the AT1R antibodies. What is the occurrence rate of this AMR? You described a high PRA and the presence of five other autoantibodies in the serum. Is it possible that these special antibodies interact with the severity of the AT1R antibody activation? How would you advise the valsartan treatment in these cases?
The case is interesting, more explanation for the special treatment.
Author Response
Thank you for your helpful feedback. Please see below for a point-by-point response:
- We have included suggestions of how we would manage a patient of this type moving forward
- We have included an estimated occurrence rate of AT1R-related AMR in other transplant types and potential relation to heart transplant
- We have included further description of potential synergism between antibody types for rejection
- We have included how we would advise management of ARB therapy post-transplant